# Factors driving the compositional diversity of *Apis mellifera* bee venom from a *Corymbia calophylla* (marri) ecosystem, Southwestern Australia

**Daniela Scaccabarozzi**[1]*, **Kenneth Dods**[2]*, **Thao T. Le**[3], **Joel P. A. Gummer**[2,3], **Michele Lussu**[4], **Lynne Milne**[1], **Tristan Campbell**[1], **Ben Pan Wafujian**[5], **Colin Priddis**[2]

**1** Research Service, ChemCentre, Resources and Chemistry Precinct, Bentley, WA, Australia, **2** Research and Innovation Division, ChemCentre, Resources and Chemistry Precinct, Bentley, WA, Australia, **3** School of Science, Edith Cowan University, Joondalup, WA, Australia, **4** Regional Institute for Floriculture (IRF), San Remo, Italy, **5** Australian Natural Biotechnology Pty. Ltd, Perth, WA, Australia

\* kdods@chemcentre.wa.gov.au (KD); daniela.scaccabarozzi@curtin.edu.au (DS)

## Abstract

Bee venom (BV) is the most valuable product harvested from honeybees ($30 - $300 USD per gram) but marginally produced in apiculture. Though widely studied and used in alternative medicine, recent efforts in BV research have focused on its therapeutic and cosmetic applications, for the treatment of degenerative and infectious diseases. The protein and peptide composition of BV is integral to its bioactivity, yet little research has investigated the ecological factors influencing the qualitative and quantitative variations in the BV composition. Bee venom from *Apis mellifera ligustica* (Apidae), collected over one flowering season of *Corymbia calophylla* (Myrtaceae; marri) was characterized to test if the protein composition and amount of BV variation between sites is influenced by i) ecological factors (temperature, relative humidity, flowering index and stage, nectar production); ii) management (nutritional supply and movement of hives); and/or iii) behavioural factors. BV samples from 25 hives across a 200 km-latitudinal range in Southwestern Australia were collected using stimulatory devices. We studied the protein composition of BV by mass spectrometry, using a bottom-up proteomics approach. Peptide identification utilised sequence homology to the *A. mellifera* reference genome, assembling a BV peptide profile representative of 99 proteins, including a number of previously uncharacterised BV proteins. Among ecological factors, BV weight and protein diversity varied by temperature and marri flowering stage but not by index, this latter suggesting that inter and intra-year flowering index should be further explored to better appreciate this influence. Site influenced BV protein diversity and weight difference in two sites. Bee behavioural response to the stimulator device impacted both the protein profile and weight, whereas management factors did not. Continued research using a combination of proteomics, and bio-ecological approaches is recommended to further understand causes of BV variation in order to standardise and improve the harvest practice and product quality attributes.

**Data Availability Statement:** All relevant data are within the manuscript and its S1 Appendix, S1, S2 Figs and S1–S4 Tables.

**Funding:** This study was funded by Western Australian Department of Primary Industries Regional Development, Export Competitiveness Grants, grant ECG007, and by the Fight Food Waste Cooperative Research Centre (CRC) grant 2.1.2. The funders provided support in the form of salaries and on-costs to authors [DS, KD, TL, JG, LM, TC, BW, CP], but did not have any additional role in the study design, data collection and analysis, decision to publish, or preparation of the manuscript. The study was led and managed by ChemCentre, a Western Australian State Government Statutory Body that provided the venom isolation and analytical chemistry and is the repository of the research record and data.

**Competing interests:** The commercial affiliation, Australian Natural Biotechnology Pty, along with funding bodies, ChemCentre and ChemCentre researchers "do not alter adherence to PLOS ONE policies on sharing data and materials.

## Introduction

Beyond the crucial pollination service provided to ecosystems and agricultural crops, bees have provided, for thousands of years, products beneficial to humans. These include honey, beeswax, pollen, venom, royal jelly, and propolis [1–3]. Bee venom (BV hereafter) has been highly valued given its broad therapeutic potential and active protein and peptide composition [4, 5]. BV is the most valuable product produced by honeybees [6], with prices varying from $30.00 USD up to $300.00 per gram, depending on the purity, composition, and / or preparation of the BV product. However, for the full commercial value to be recognised, uniform production of BV and a chemical composition certification need to be delivered reliably to the marketplace. Standardising venom composition would assist product formulation and advantage BV research and applications in medicinal fields by consistently and clearly presenting the causal agents responsible for the therapeutic effect.

Venom produced by honeybees (apitoxin) is an assorted and synergistic mix of biochemically and pharmacologically active proteins, including polypeptides (melittin, apamin, and mast cell degranulating peptide), amines (histamine, serotonin, dopamine, and norepinephrine), and enzymes (phospholipase, hyaluronidase, histidine decarboxylase) [7]. Bee venom use in alternative medicine dates back more than 5000 years [2]. Yet, in the last decade apitoxin, from *Apis mellifera* (Apideae) has been widely studied and increasingly sought for its therapeutic features and pharmaceutical applications in the alternative treatment and prevention of disparate degenerative and infectious diseases, such as cancer, HIV, multiple sclerosis, arthritis, bursitis, tendonitis, dissolution of scar tissue, herpes zoster, joint diseases and rheumatoid arthritis, Lyme disease, and osteoarthritis [4, 5]. A newly expanding field of application is the personal care and the cosmetics industry [8]. Melittin and apamin, the two main active peptide components of BV, are the most commonly investigated peptides reported as used in research and medicinal fields [9]. Both are well regarded for their anti-inflammatory, -nociceptive, cytotoxic action against cancer cells [10, 11]. Melittin especially, is known for its powerful anti-microbial action [11]. Since the beginning of the 21$^{st}$ century, the emergence of different separation techniques followed by mass spectrometry has enhanced the identification of more protein/peptide components of BV [12, 13], including the various isomers and enantiomers of melittin, phospholipase A2, apamine, and mast cell degranulating peptide. More recently, these techniques have also been applied for the identification of new proteins and peptides (e.g., toxins and allergens) possessing biological activities [13].

Icarapin (Api m 10) is the protein responsible for the major allergenic response to bee stings, followed by the well-characterized phospholipase A2 (Api m 1; [14], Api m 4 (melittin) and the Api m 6, the latter showing 42% IgE (Immunoglobulin E) reaction [15]. Bees, wasps and ants (Hymenoptera) produce venom for defence against potential predators and intruders. The venom, which is synthesized by the venom gland (Dofour's gland), is stored in the venom reservoir where both acidic and basic secretions are compiled and mixed [16]. Bee venom production increases during the first two weeks of the adult worker's life and reaches a maximum when the worker bee turns into a hive defender and forager [17, 18]. Bee venom production is reported to be highest during the summer months, corresponding with peak hive activity [4, 19]. It consists of 88% water [4, 20] and dries rapidly with air contact [21]. Under optimum conditions, 1 g of venom is produced by 10,000 worker bees over ~100 minutes [6, 22] and between 3–4 μl of venom is expected to be in a single bee worker [6]. Generally, only a fraction of bee venom, 0.5–1.0 μL, can be obtained from a stinging event from which 0.1 μg of dry venom can be isolated [6].

Bee venom secretion and composition are influenced by a multitude of biological and hive management factors including honey bee intraspecific variation, age of bees, colony strength,

defence behaviour, nutritional supply, season, and method of venom collection [22–27]. Global comparison of venom electrophoretic profiles of 25 different hymenopteran species revealed that the protein patterns strongly differ from one species to another [28]. When comparing the venom of the Carniolan honeybee and the Italian honeybee *(A. mellifera carnica and A. mellifera ligustica)* to the venom of the Africanized honeybee hybrid, their protein compounds differed both in presence and abundance [29]. Next to inter-species variation, intra-specific variability can be as marked as showing different venom between bees of the same population of the same age [30]. Consistent differences in the composition of one venom fraction were observed between honeybees performing different tasks [31]. For example, relative percentages of two key cytolytic peptides in BV, namely melittin and apamin, differ between queens and workers, younger and older workers, nurses, guards and foragers [32, 33]. Notably, the composition of the worker bee venom varies as a function of time and is probably related to the transition from 'house bee activity' to 'field bee activity' [6]. Even nutritional supply, such as pollen substitutes, has been shown to increase major constituents of BV [27].

Whilst previous studies have explored the impact of biological factors influencing bee venom protein nature and amount, little research has been conducted to clarify the causes of bee venom variation in respect to the ecological context. Classical works have linked the protein composition variance of BV to annual seasonality [34, 35]. Also, intraspecific diversity of melittin and phospholipase A2 in venom from honeybees was associated by Ferreira Junior et al. [21] with climatic and seasonal factors. However, the study didn't link information to flowering composition over the study period (one year), so that seasonal variation might also be a reflection of the flowering vegetation throughout the seasons. According to Danneels et al. [30], the seasonal variation between summer and winter worker bees, using proteomics techniques, showed the antigen 5-like gene expression in venom gland tissue solely in winter bees. Additionally, six of the 34 venom toxins produced by worker bees in winter and/or summer were not detected in queen bees (group XV phospholipase A2, 5′-nucleotidase, carboxylesterase, serine protease snake, serpin 3, and c-type lectin).

This study aims to characterize the bee venom released by bees during the flowering season of *Corymbia calophylla* (Myrtaceae; marri), and to elucidate the ecological and biological factors that influence BV proteins' composition and the amount produced. Due to the species abundance, marri honey is the most produced mono-floral honey type of Southwestern Australia. It is also greatly appreciated and commercialised for its organoleptic properties. Although Western Australian BV is scarcely harvested, BV obtained from flowering marri can significantly enhance apiculture profitability, if optimally produced and characterised.

Based on the biological assumption that target bees belonged to the species *Apis mellifera ligustica*, and come from the same breeding program, the study aims to quantify the weight of BV released by bees per hive and identify the BV proteins using proteomics to test if bee venom proteins' composition and weight: i) vary between sites; ii) are influenced by ecological factors (temperature (˚C), relative humidity (%), flowering index and season, nectar production); iii) are affected by hives management practices (nutritional supply and historical movement) and; iv) behavioural response (biological factor). Furthermore, to appreciate the qualitative variance in relative abundances of key proteins possessing diverse bioactivity, more specifically, differences between melittin and apamin, and between icarapin and allergen, were tested as a function of the flowering season and behavioural response respectively.

Indirectly, by assembling a protein profile for comparison, the study proposes a fundamental tool for establishing the product certification, vital for BV value determination and commercialisation.

## Materials and methods

### Study sites and features

Twenty-five beekeeper-managed honeybee colonies in five study sites in Southwestern Australia's native Eucalypt forests were located on the Darling scarp, a low escarpment running north–south and to the east of the Swan Coastal Plain and Perth, the capital city of Western Australia. The sites, ranging from north to south over a latitudinal extent of approximately 200 km, were: Chittering (-31.55 S, 116.05 E), Chidlow (-31.85 S, 116.30 E), Hovea (S -31.87, E 116.11), Byford (S-32.26, E 116.05), and Harvey (S -33.04, E 115.96) (Fig 1). Permission for placing apiaries on crown land in Byford and Harvey was issued by Department of Biodiversity, Conservation and Attractions (DBCA), whilst single owners gave permission for sites on private land (Chittering, Chidlow, Hovea). The study was conducted during the flowering season of *Corymbia calophylla* (Myrtaceae; marri). Formerly unpublished research led by the research team on pollen and nectar composition, demonstrates that a bee diet based on marri as floral source, meets the dietary maintenance requirements for honeybee health and nutrition and did not involve endangered or protected species. Palynological analysis of honey and pollen traps collected over the sampling season, was conducted to determine the major flowering component of the vegetation during the sampling time, and therefore the principal bee foraging diet and honey composition at the time of the venom collection (Fig 2A and 2B; S1 Appendix; S1 Table). Bee colonies in hives contained the same number of frames and the number of bees ranged from 35,000 to 45,000 per colony (hive).

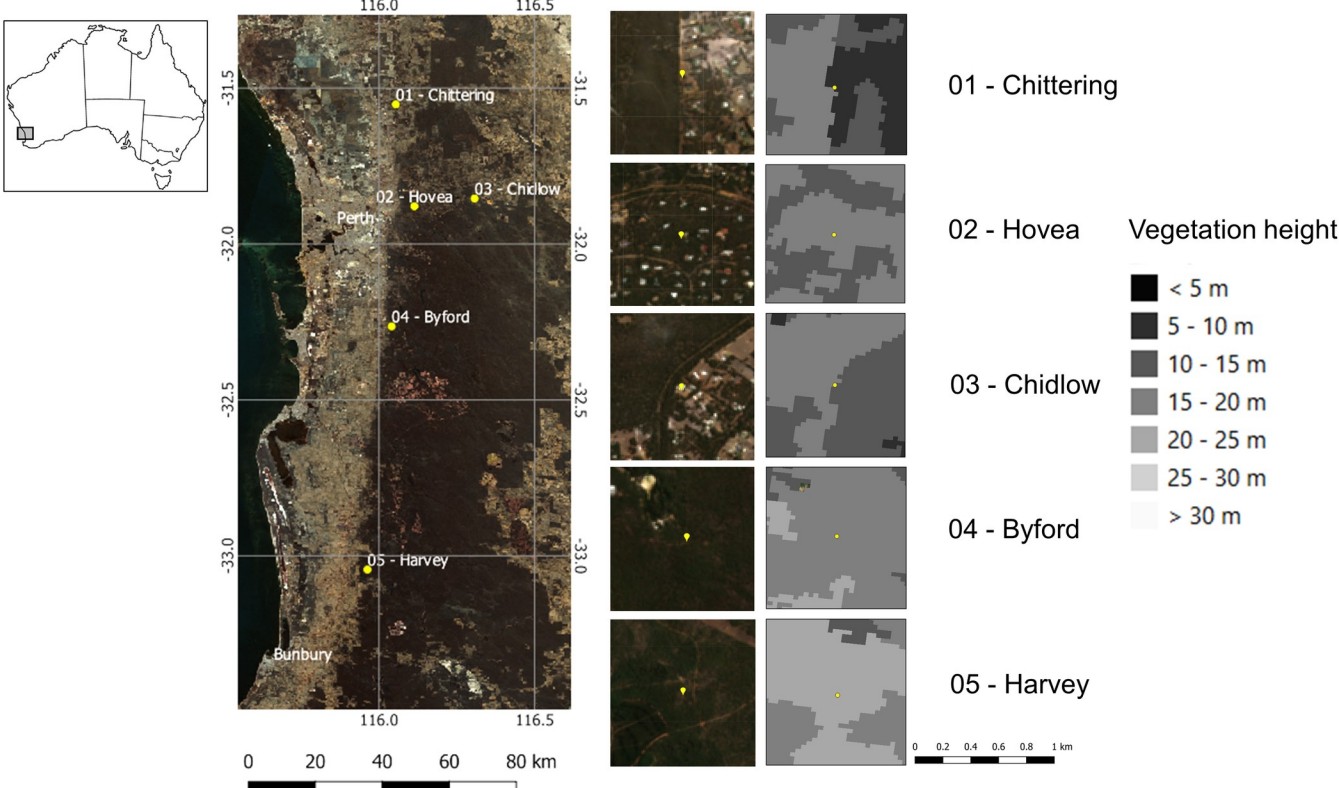

**Fig 1. Sampling sites and canopy height.** Distribution of the five study sites in the Eucalypt native forest according to ABARES [36] in South Western Australia. Canopy data (vegetation height) derived from Scarth [37]. Satellite imagery are Copernicus Sentinel Data [38].

## Bee venom sampling and storage

Sampling was conducted from 21 January to 6 March 2020 from five sites (Fig 1). In each site there were five hives and the sampling was carried out twice over the flowering season, one in the first half and one in the second half. We used electrical stimulating devices for collecting bee venom (C-J 201 ® Cheongjin Tech, Korea), formed by a glass plate overlaid with metal filaments, fed by a power unit combined with an electrical pulse system [39] and placed at the entrance of the hive (Fig 2C). The electrified wires provoke a low voltage minimal shock to the bees that respond by stinging the surface, remaining active and alive [21]. Plates were externally placed on five hives per site to conduct one-hour venom sampling between 10.00 am and 13.30 pm, with the electrical voltage set at 16.8 Volt [22]. The venom dried rapidly on the glass plates. Plates were packed in separate boxes to avoid contamination during transport to the ChemCentre venom specialist laboratory (corner of Manning Road and Townsing Drive, Bentley, Western Australia). In the laboratory the BV was scraped from the glass plate with a razor blade and transferred to a dark glass container under ambient conditions. Safety practices were employed to conduct bee venom storage. Special equipment was worn to avoid bio-contamination (gas-mask, protective glasses, disposable rubber gloves, a lab coat and boots). Scraping and handling practices were conducted in a low velocity laminar air flow cupboard to adequately contain and protect against venom particulate dispersal. Samples were stored at -20˚C.

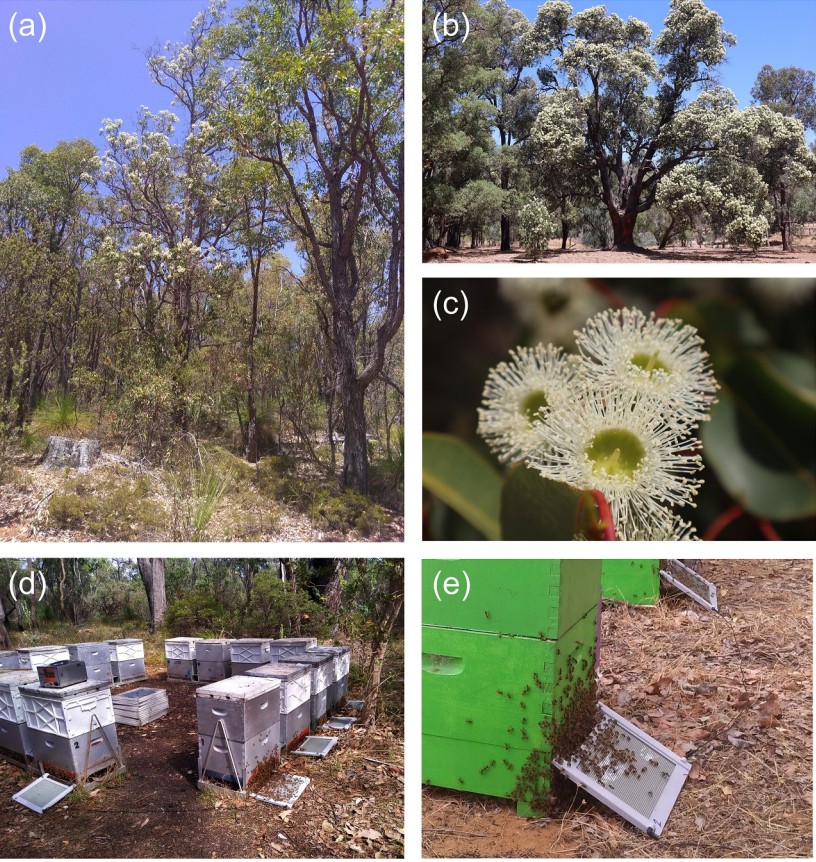

**Fig 2. Bee venom sampling during the flowering of *Corymbia calophylla* (marri).** (a) Marri woodland ecosystem. (b) *C. calophylla* in peak flower, (c) *C. calophylla* in flower. (d-e) bee venom harvest using stimulating glass plates at the front of the hives.

## Proteomics

To prepare BV for analysis, the dried venom was dissolved in 40 mM triethylammonium bicarbonate (TEAB) to a final concentration of 40 mg.mL-1. The bee venom solution was mixed and sonicated at room temperature. Samples were subsequently diluted to 10 mg.mL-1 with TEAB. Extraneous particulate material in the venom (bee faeces, dust, pollen, honey, wax) was removed by centrifugation at 20,000 g / 5 min and the supernatant (venom) was transferred to a fresh polypropylene tube. The solubilised protein (400 μL) was prepared for tryptic digestion by the addition of 2 μL 0.5 M dithiothreitol (DTT), 2 μL 1% ProteaseMAX™ (trypsin enhancer; Promega Corporation, Australia) and incubated at 56°C for 20 min. To the warmed solution 4 μL of 0.5 M iodoacetamide (IAA) was added, followed by incubation at 37°C for 30 min in the dark. An additional 2 μL of 1% ProteaseMAX™ and 10 μL 0.4 μg.mL-1 trypsin was added and the digest allowed to proceed over 16 hours with incubation at 37°C. Subsequently, the entire volume was filtered with a 0.45 μm UltraFree MC centrifuge filter (Merck-Millipore, Germany) and the filtered solution flow through collected by centrifugation at 14,000 rpm / 10°C / 10 min. The filtered sample was then dried by vacuum concentration. In preparation for LC-MS/MS analysis, the sample was redissolved in 2% acetonitrile (0.1% formic acid). The digests were analysed by LC-MS/MS by injecting 25 μL into a Thermo Vanquish UPLC (Thermo Fisher Scientific), coupled to a Q-Exactive Plus Orbitrap Mass Spectrometer (Thermo Fisher Scientific). The LC was equipped with an ACQUITY UPLC Peptide BEH C18 (2.1 × 150 mm, 300 Å, 1.7 μm; with column guard) analytical column, held at 40°C, using mobile phases 0.1% v/v formic acid (A) and 99.9% acetonitrile (0.1% v/v formic acid, B) at a flow rate of 300 μL.min-1 and optimal mobile phase ratios selected to enhance peptide separation. The MS was operated with a full scan-ddMS^2 acquisition, using a mass range of m/z 350–2,000, an AGC target of 3e6 with a resolution of 70,000 FWHM (m/z 200) and using a maximum ion injection time of 50 ms. Product ion (dd-MS2) spectra were acquired with an AGC target of 3e6 at a resolution of 17,500.

Protein identification and quantification was obtained by processing raw LC-MS/MS data on Proteome Discoverer™ (PD) version 2.2 (Thermo Fisher Scientific). Precursor peptides were selected from MS1 spectra, requiring a mass tolerance of less than 10 ppm. Product ions were identified within a mass tolerance of 0.05 Da. Protein identification required a maximum of two missed cleavage sites and a minimum and maximum peptide length of six and 144, respectively. Sequences were identified by comparison to the *A. mellifera* reference proteome (downloaded from Uniprot database); requiring a total number of peptide spectra to match with a false discovery rate (FDR) of 0.01 (strict) and 0.05 (relaxed). Peptides were filtered such that only high confidence peptide identities with a minimum peptide length of six and with at least a single peptide were utilised. Protein was grouped by following the strict parsimony principle. Untargeted label-free quantification, particularly precursor ion quantification of protein was carried out by processing the dataset using default processing and consensus workflows available within PD 2.2. The protein abundance was computed as the sum of the peptide group abundances associated with that protein.

## Ecological factors driving BV weight and protein composition

To determine ecological factors affecting the bee venom quantity over the marri flowering season, a suite of variables was monitored. Climatic factors, temperature (°C) and relative humidity (%), were monitored using *in situ* meteorological stations (Davis Instruments, Vantage Pro 2, Australia). The ecological factors, marri nectar production, flowering index and stage, were obtained by direct sampling, satellite images and observational records, respectively.

To quantify nectar quantity produced by marri trees at the field sites, nectar sampling was conducted by sampling flowers using a ground-based tool [40]. As nectar volume and texture visibly varies between the first and second half of the marri flowering season (Tristan

Campbell, personal comment) from a viscous to a liquid form, extraction was conducted initially by using a 1 mL micropipette with a known amount of water [41], washing five flowers per sample. We measured the extra volume related to the known volume of added water. Once the nectar volume increased visibly over the season and had a liquid form, it was extracted from a single flower using 20 µl micro capillary tubes (Drummond Microcaps, Broomall, PA, USA). The volume of nectar extracted by capillary tubes was estimated by measuring the length of the column of liquid along the tube [42] using a digital caliper. A total of two measurements on five plants per site were made. The flowering index data for collection sites was based on satellite images using Sentinel-3 satellite remote sensing platform. To develop a reliable temporal curve, a moving average of the maximum calculated marri flowering index by the ratio of Band 6 to Band 1 (MFI; according to Campbell and Fearns [43]) over a 14-day period was used. This is similar to the maximum window approach commonly used to calculate other vegetation indices, such as the 16-day Normalised Difference Vegetation Index (NDVI) product produced by NASA [44].

Flowering period was described based on observations from the start to the end of the marri flowering season, assigning 'first' or 'second' flowering stage, before or after the mid-date, over the whole flowering range.

## Hives' management and behavioural factors influencing BV

Management practices such as nutritional supply and historical movement of hives were recorded by interviewing the beekeepers. Nutritional supplements provided to bees at the beginning of the spring flowering season (between August and September) included: none, marri (*Corymbia calophylla*) pollen, and unspecified pollen plus soy flour and sugar syrup. Historical hives' movement was classified as follows: stationary (permanently placed in the marri-jarrah forest), mainly stationary (the predominance of the year the hives are placed in the marri-Jarrah forest), seasonally moved (moved periodically over the year based on flowering seasons). Bee behaviour was expressed into two classes based on field observation and video recordings of bees stinging against the stimulator device [45]. The number of bees interfacing the plates was recorded twice over three-minute sessions. For each observational period we assigned 'docile 'and 'active' categories, based on the number of bees on the bee venom plate, (<80) and (>80), respectively.

## Statistical analysis

To test the hypothesis that venom dried weight differs between variables, we performed a permutational multivariate ANOVA (PERMANOVA; [46]) by testing the simultaneous response of all the measured variables. We used Euclidean distance on standardised data, 999 unrestricted permutations of raw data using correct permutable units; the pair-wise tests were corrected for multiple comparison. To determine the effect of variables on venom weight, we used generalized linear models (GLMs) with gaussian distribution because our dependent variables were mainly continuous. The model selection was performed using the "glm" function using the package vegan in R Studio (Version 3.3.2), covering all possible combinations. The best fitting model was chosen based on the Akaike's information criterion corrected for small sample sizes (AICc). The relative importance of each of the variables selected by the best GLM model was evaluated performing hierarchical partitioning. Patterns of protein variation were summarized by principal component analysis (PCA). The disparity in proteins composition for samples collected in five sites was tested by similarity analyses. T-Test and Kruskal-Wallis H-test were applied to identify the relevance of behaviour and flowering stage on the four target proteins (apamin, melittin, allergen, icarapin). Prior to analysis, data were log transformed.

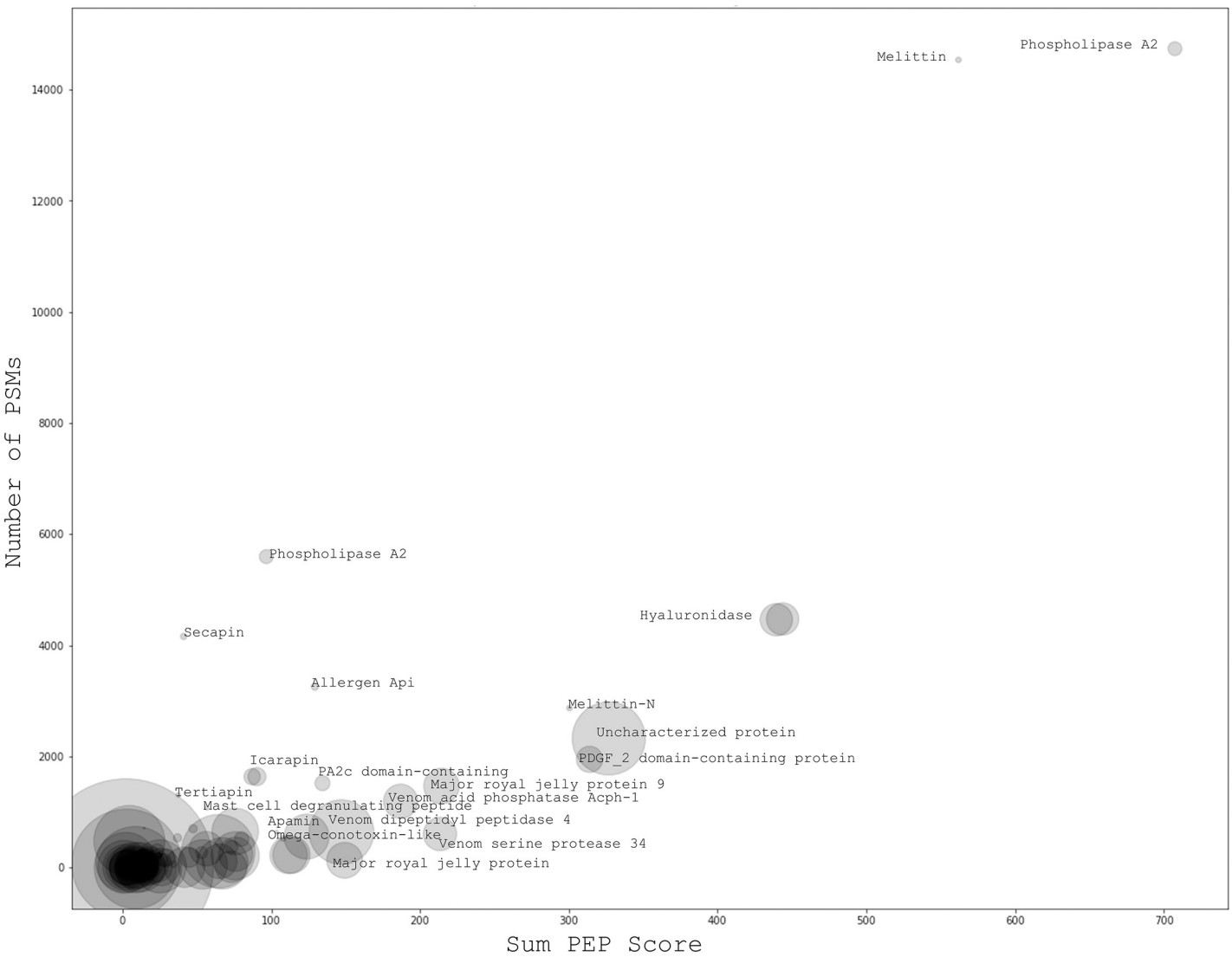

**Fig 3. The major contributing compositional proteins of bee venom, as determined by LC-MS/MS.** Higher abundant proteins represented by the combined peptide match score (Sum PEP Score) and number of peptide spectrum matches (PSMs). Bubble size indicates molecular weight; S3 Table presents the full list of proteins with relative names and molecular weight.

## Results

### Protein composition of bee venom

To determine if the protein composition varied in relation to each of the studied variables, the BV proteins were measured by LC-MS/MS. A considerable contribution of phospholipase and melittin amongst other known venom proteins was seen. A number of uncharacterized proteins were also well-represented in the BV (Fig 3, S2 and S3 Tables).

### Ecological factors driving the BV weight and protein composition

*Corymbia calophylla* (marri) represented the principal bee nutritional component at the time of the venom collection (Fig 2A and 2B; S1 Appendix; S1 Table). The highest weight of venom released by bees on the stimulating devices came from Harvey (mean± SD, 0.072 g ± 0.035),

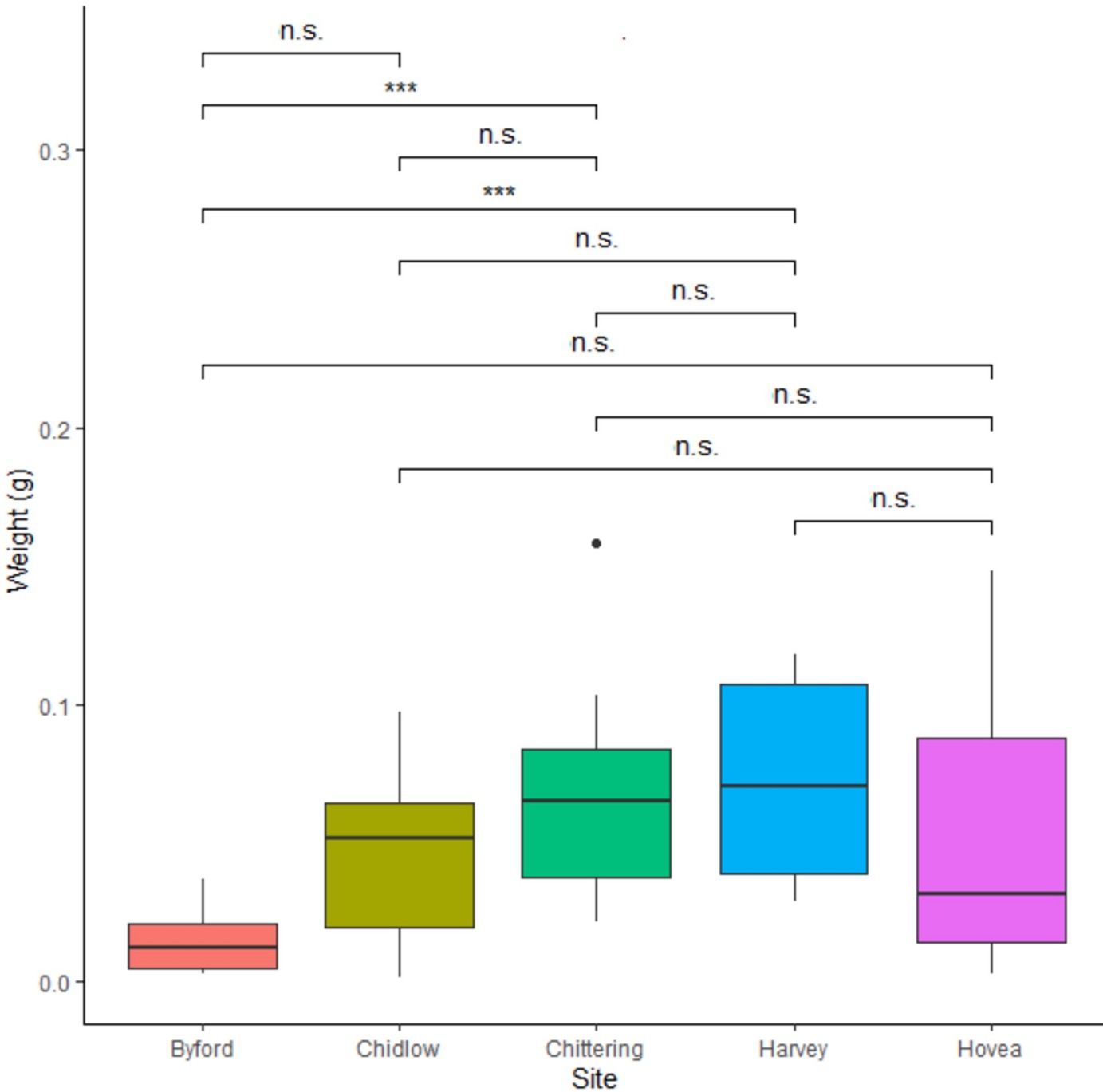

**Fig 4. Weight of bee venom samples per study sites.** Weight in g; medians and standard deviation are shown.

whereas the lowest was from Byford (mean± SD, 0.014 g ± 0.011) (Fig 4; S4 Table). Variation of BV's weight among sites was found only between Chittering and Byford (p = 0.013) and between Harvey and Byford (p = 0.0084). The principal component analyses based on site (S1 Fig) indicated that the first two axes could explain 56.1% of the protein variation between samples, 42.5% by axis 1 and 13.6% by axis 2. The most compact cluster was associated with the Chittering site, while BVs from the Byford site showed the most dispersed distributional

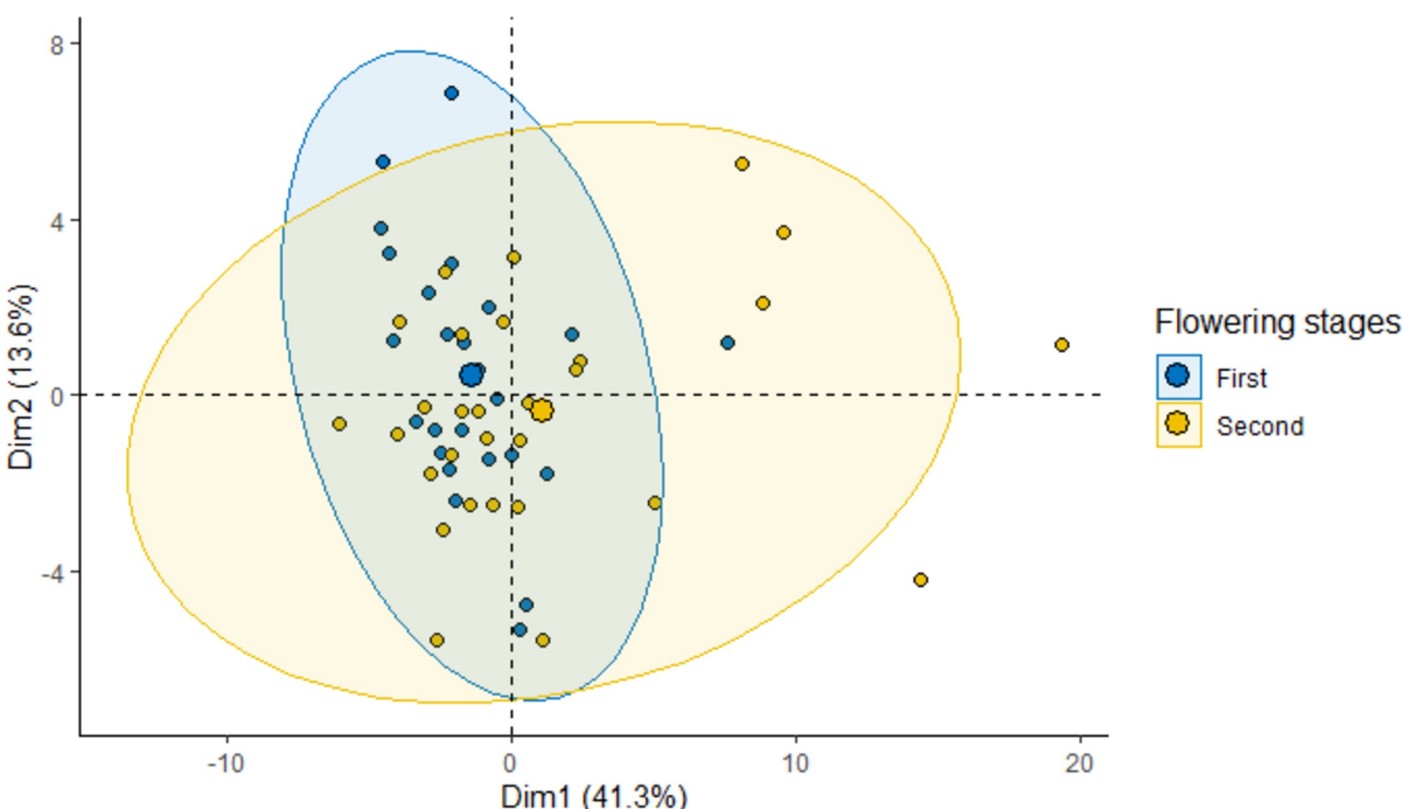

**Fig 5. 2D PCA-plot of protein profile in function of marri flowering season.** Biplot of the first and second axes obtained from the principal component analysis (95% confidence ellipses are shown) showing dependencies of protein profile between first and second marri flowering stage. Points represent venom samples (n = 55).

area. In respect to weight, we found a significant negative relationship with temperature ($R^2$ = 0.0815, p > 0.05), whilst relative humidity, marri flowering index and nectar production were not related. Temperatures during the sampling season ranged from 20.7˚C to 30.7˚C, and relative humidity varied from 36% to 71%. Weights significantly varied over the flowering period with higher values corresponding to the first flowering stage, $t_{df}$ = 2.33$_{33.976}$, p = 0.026.

Among ecological variables, the results of the similarity test on proteins showed that protein composition was influenced mainly by sites (p = 0.001) and temperature (p = 0.002). Other ecological variables such as relative humidity, marri flowering index and nectar production didn't show any effect on the protein composition of BV. The principal component analyses performed on all data (Fig 5) indicated that the first two axes could explain 54.9% of the protein variation between samples, 41.3% by axis 1 and 13.6% by axis 2. We found a clear discrimination between flowering ranges (first and second), with a strong cluster especially for the first flowering stage, corresponding to the first half of the marri flowering season. Relative abundance of major proteins, melittin and apamin, did not vary over the marri flowering season.

## Management and behavioural factors influencing BV weight and composition

Among the management (nutritional supplement, and historical movement of hives) and behavioural factors influencing the BV quantity, only bee behaviour, as response to the stimulating device, was related to BV weight. Both nutritional supply and historical movement of

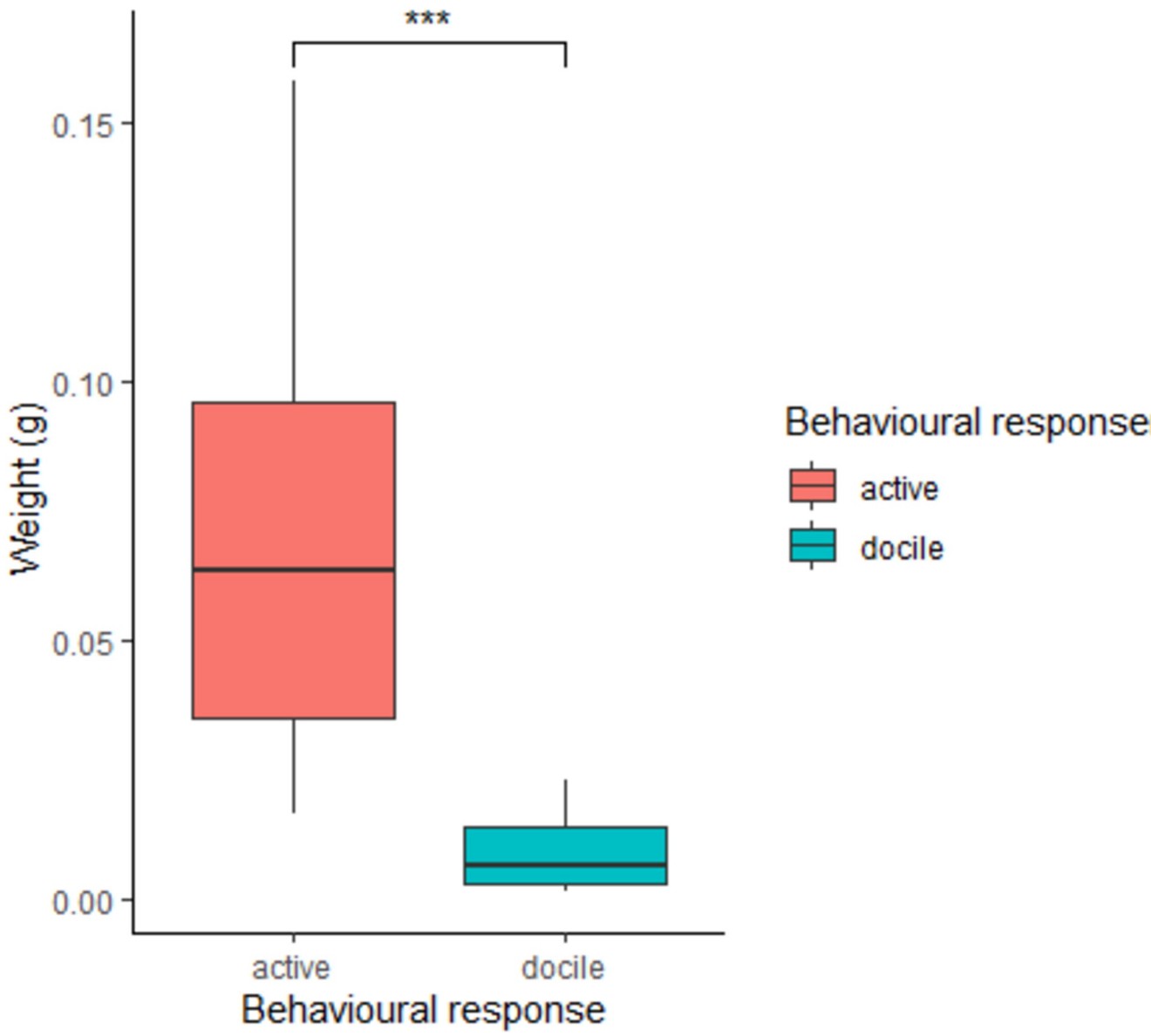

**Fig 6. Weights of bee venom samples per behavioural reaction to the stimulating devices.** 'Docile' and 'active' corresponded to number of bees interfacing the stimulating device, <80 and >80 respectively. Weight in g; medians and standard deviation are shown.

hives were not linked to BV weight. In particular, lowest BV weight was associated with docile bees (mean±SD, 0.0091 g ± 0.0068), while highest BV weight was related to active bees (mean ±SD, 0.0672 g ± 0.0362) (Fig 6; S4 Table). Provision of different nutritional supplies before commencing the foraging season, produced the following weight averages for BV samples: none, (mean±SD, 0.0410 g ± 0.0519), unspecified pollen plus soy flour and sugar syrup (mean ±SD, 0.0471g ± 0.0532), and marri pollen (mean±SD, 0.0471 g ± 0.0310). The results of the similarity test on proteins variance showed that protein composition was influenced mainly by behaviour (p = 0.008), whereas other management factors were not linked to protein variability. The principal component analyses performed on all data (S2 Fig) indicated that the first two axes could explain 56.1% of the protein variation between samples, 42.5% by axis 1 and 13.6% by axis 2. Data analysis showed two distinct clusters, corresponding to the docile and

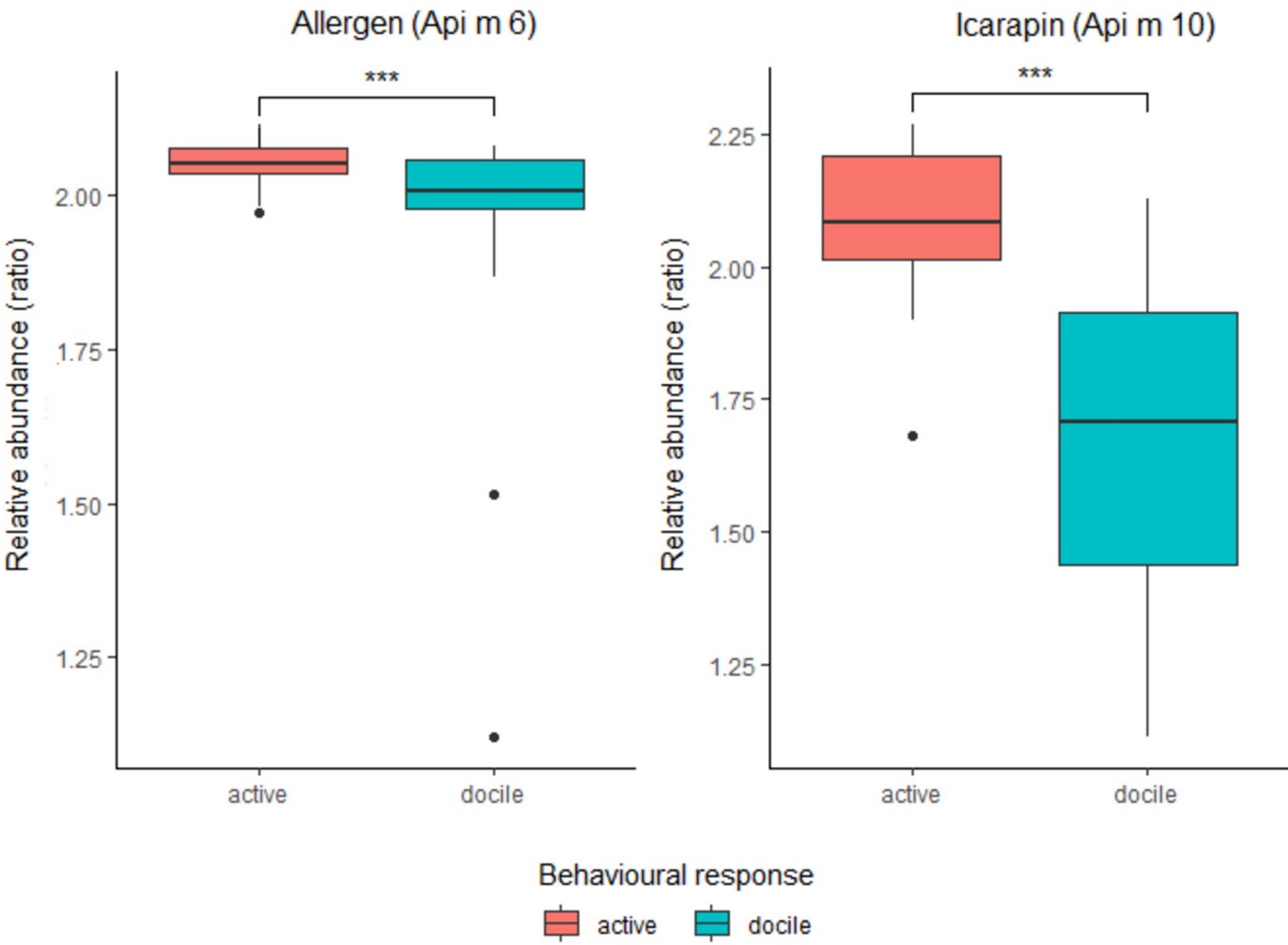

**Fig 7. Relative abundance of allergen (Api m 6) and icarapin (Api m 10) proteins per behavioural reaction to the stimulating devices.** Relative abundance is a ratio of protein abundance in a sample to that in a control, obtained by untargeted label-free quantification. 'Docile' and 'active' corresponded to number of bees interfacing the stimulating device, <80 and >80, respectively. Weight in g; medians and standard deviation are shown.

active bees, respectively. Among the proteins associated to the allergenic response, we found that Allergen and Icarapin varied depending on behaviour reaction to the stimulator device ($t_{df} = 2.33_{14.024}$, p = 0.035; $t_{df} = 2.5833$, $_{39.414}$, p = 0.013) (Fig 7).

Among all candidate models for weight as a dependent variable, the one including 'temperature', 'relative humidity', 'behaviour', 'nutritional supply' and 'movement' had the lowest AICc (117.94). With regard to proteins composition, the best fitting model that included 'site', 'temperature' and 'behaviour' was found to be statistically supported in defining variation of protein composition among samples (p<0.001).

## Discussion

The protein and peptide composition of BVs from colonies of honeybees (*A. mellifera ligustica*) principally foraging on *C. calophylla* (marri) (S1 Table) in South Western Australia, was examined here, employing an untargeted proteomics approach to characterise the proteins of BV and identify causes of its molecular variance.

These data describe changes to the protein composition of BV, from a profile of 99 proteins, including established components of BV [13, 47–49] (S2 Table) with up to two-thirds of the described proteins newly reported in this study. Consistent with literature accounts of BV proteins, a considerable contribution of phospholipase and melittin [13, 47, 49], amongst other known venom proteins, (Fig 3; S2 Table) was also found. The relative composition of proteins endogenous to Australian marri bee venom (S2 Table) has provided a comprehensive characterisation toward meeting international ISO 17025 standards. This will allow product certification and protect this highly valued commodity. Bee venom standardisation to date has been developed by Lee et al. [50], demonstrating a purification process that separates allergens from other active components. Alternatively, Orbitrap LC-MS/MS results for marri bee venom are presented using a global proteomics approach, including the allergenic components allowing evaluation of synergistic effects and natural component peptides ratios.

Here we provide an understanding of BV quantity (weight) harvested per hive, with implications for developing cost-effective strategies for BV harvest.

The variation of peptide profile and BV weight considered multiple ecological factors, including bee hive management practices and insect behaviour, with data indicative of important correlations with the potential to assist venom production compositional consistency.

## Factors influencing BV weight

Average of BVs' weight harvested per hive was 0.051 g (± 0.040 SD). The secretion of BV per hive is according to the reference value obtained by 10,000 worker bees under optimal conditions of 1 g per 20 hives [6, 22] within a shorter yield timeframe (60 minutes instead of 100 minutes). BV weight diversity between hives of the same site appears to be most pronounced in the Hovea site and least pronounced in the Byford site (Fig 4).

**Ecological factors.** When considering ecological context, the primary driver that explained the variance of BV weight was temperature, while the geographical distribution (site) was a determinant for the variance between two sites. Bee venom from the Byford site, mid-located over the latitudinal range, was significantly different from BV coming from the two extreme-sites, Chittering (northern-end) and Harvey (southern-end). This geographical evidence suggests that exogenic site factors definitely warrant further work to enable effective control of product quality and to design cost-effective strategies for BV harvest.

Interestingly, Byford, in accordance with the canopy classification reported in Fig 1, presented the most uniform forest height (between 15–20 m) compared to other sites (see Fig 1), and the lowest BV harvested. According to the evidence that differences in plant height are related to the forest maturity grade [51], Byford site fits in a young forest context. However, though not addressed here, it would be of relevance for future investigation to test if younger forests lead to lower BV harvests than older forests.

With higher temperature BV weight declined. High temperature can be detrimental to bees' activity and fitness in and out of the colonies, stressing energy consumption to maintain constant hive temperature within the optimal range from 33 to 36°C [52, 53]. Relative humidity is another vital abiotic factor for bee development. Below 50% it hinders egg hatching [54], affecting bee biological development inside the colony. Foraging activity has also been shown to be negatively impacted by temperature [55], but not markedly affected by relative humidity [56]. This latter evidence supports the absence of a relationship between BV weight and relative humidity, and to the negative impact on BV weight with higher temperatures.

Neither flowering index or nectar production was clearly associated with BVs weight. However, to tease apart the effect of these environmental variables and define such ecological trends, a multiple-year-study is certainly required. On the other hand, BV weight was a

function of the marri flowering season, named here as first and second flowering stage. In particular, the beginning of the marri flowering season lead to more copious BVs, indicating a higher secretion of venom in the first marri flowering half when bees predominantly harvest pollen rather than nectar (Tristan Campbell, personal observation). Our finding well-meets the expectation that continued nutritional supplement with pollen substitutes increases the BV quantity [27] and quality [57]; the latter does not differ by comparing bees with a pollen-based diet to bees with a natural pollen grains diet [58].

**Management and behavioural factors.** In respect to biological features, a compelling behavioural factor was revealed by the association between the behavioural response to the stimulating device (docile and active bees) and apitoxin weight. In fact, the overall quantity of BV released by bees relies on the alarm pheromone secretion that induces other bees to aggressively react by stinging [6]. Thus, the aggressive degree of bee colonies is determined by the number and frequency of stings on a given object that causes the attack [45]. We also underline that the best fitting model, for performing the statistical analysis treating weight as dependant variable, included 'nutritional supply' and 'historical hives' movement'. This suggests that hives' management factors might indirectly interact with the behavioural response, resulting in a varied bee defensive reaction. Although the hives' management factor didn't show a relationship with BV weight, the interaction with bee behavioural attitude needs further exploration. As multiple abiotic and biotic variables influenced the harvested quantity of bee venom per hive, these factors should be critical future points of assessment for improving BV harvest by commercial beekeepers.

## Factors driving the diversity of BV protein profile

Ecological factors that were associated with BV weight (site, temperature, flowering stage), also influenced the peptide composition and diversity. As per BV weight, amongst the biological and management causes of BV peptide diversity, bee behaviour clearly influenced the compositional variation.

**Ecological factors.** Temperature and site primarily influenced the protein compositional variance, along with the marri flowering stage (first and second; Fig 5).

Former studies by Ferreira et al. [21] and Danneels et al. [35] recognised seasonally-based (over a single year) causes of BV protein variation, but didn't find any specific relationship with climatic factors such as temperature and relative humidity. Instead, conforming to our study, temperature appears to be a determinant climatic variable able to assure a stable BV protein profile. As Ferreira et al. [21] and Danneels et al. [35] have monitored a single hive in one site, they underlined the necessity to examining multiple sites and hives to better explain the effect of climatic factors on BV protein profile and to confirm such an ecological trend. Therefore, this incongruity between the findings may be associated to a limited sample size of the former studies.

Geographical range partly explained the dissimilarity in protein composition (S1 Fig). Most of the former studies have investigated causes of BV molecular variation in a single site [21]; however, based on evidence related to other animal venoms (scorpions, snakes), it is expected that biotope, as precise geographical localization, would be one of the most influencing causes of intraspecific variation of venom [59–61]. According to the argument raised for BV weight, site factors should be carefully evaluated prior to setting up a BV harvest in order to secure a given or desired protein profile. Bee venom samples showed clear evidence of protein variance being driven by geographical location (S1 Fig). This apparently reflected the canopy height heterogeneity (Fig 1). The strongest cluster, with a lowest diversity of protein profile between BVs, was associated with the Chittering site that also presented the most varied pattern in

terms of canopy level height (Fig 1), whilst the most dispersed cluster was shown by the Byford site, characterized by the most uniform canopy height (Fig 1).

Like for BV weight, nectar production and flowering index did not impact the venom protein composition. By contrast, flowering season showed two distinct composition patterns corresponding to the first and second marri flowering stages (first and second half of the season), meeting the expectation that seasonal factors induce a change in the BV protein profile [21]. This latter evidence suggests that inter- and intra-year flowering index should be further explored to better understand its influence on BV composition. Surprisingly, key proteins such as melittin and apamin didn't vary significantly between the first and second flowering stage, demonstrating that protein variance over the flowering season influences only certain or specific functional groups of proteins. Thus, depending on the study aim, further investigation that incorporates flowering index and / or season as potential influencing factors on BV protein profile, would require to prior select target proteins, or biological functional groups of proteins, for isolating precise compositional patterns. Bee venom major components such as melittin and $PLA_2$ are expected to vary as a function of the season of the year [21, 33]. However, we underline that such diversity throughout the year can likewise be associated with a variation in the vegetation composition, determining different feeding sources for bees.

**Management and behavioural factors.** Interestingly, among biological factors, behavioural response shed light on determining differences in the BV protein composition, whilst nutritional supplement and historical movement of hives were irrelevant in drawing any kind of trend. Bee venom samples from active bees that reacted intensively to the stimulating devices had a more comprehensive protein variety. This finding may be explained as a result of health state, or strength level, of the bee colony that is linked to several phenomena including parasitism, disease or agricultural chemical exposure, foraging activity and diet [62–65], the latter particularly determining the production of BV [27]. We also underline that changes in gene expression in correspondence to alarm response, which provokes aggression by bees, can regulate the behavioural reaction [66]. In short, a behavioural reaction may be a result of a genetic asset and its expression modality. Curiously, allergen and icarapin, are among the principal polypeptides which have been shown to cause the allergenic response [14, 15]. These varied in terms of relative abundance depending on the degree of aggressive behaviour demonstrated by the bees. Moreover, specifically, higher relative abundance of these two allergenic peptides corresponded to bees that most actively responded to the stimulating device.

## Conclusions

Our study presents an advanced exploration of the BV proteome, combined with both ecological and biological information for data interpretation to dissect the causes of BV compositional variance. For the first time, we characterized BV produced by *Apis mellifera* in the South Western Australian commercial beekeeping region of the *Corymbia calophylla* (marri) ecosystem. The relevant ecological and biological relationships with BV weight and protein profile (relative abundance and composition) were determined as driving factors of BV compositional diversity and included temperature, site, marri flowering season and bee behaviour. Future research on ecological and behavioural features to better elucidate the causes of BV variance is encouraged. An environmental approach, combined with biological indicators of the bee colony state (strength / health), was explored. In fact, a better comprehension of biotic and abiotic factors influencing BV variance can lead to a standardization of BV harvest by apiarists to secure the locally produced BV a reliable place in the global market. The study provides a fundamental tool for establishing next generation NATA and ISO chemical analyses certification programs for BV. Future development will see these measured attributes become a valuable

inclusion to such certification, enabling the quantitative measure of bioactivities and other quality traits of specific BVs. This capability would also address the growing demand for BV in clinical and therapeutic fields, with substantial benefit to both human health and the beekeeping industry/primary producers.

## Supporting information

**S1 Fig. 2D PCA-plot of protein profile in function of sites.** Biplot of the first and second axes obtained from the principal component analysis (95% confidence ellipses are shown) showing dependencies of protein profile between sampling sites. Points represent venom samples (n = 55).
(TIFF)

**S2 Fig. 2D PCA-plot of protein profile in function of behavioural response.** Biplot of the first and second axes obtained from the principal component analysis (95% confidence ellipses are shown) showing dependencies between active and docile bees. Points represent venom samples (n = 55).
(TIFF)

**S1 Table. Composition of pollen from honey samples collected in field sites at the end of the flowering season of *Corymbia calophylla* (Myrtaceae).** Pollen percent occurrence was calculated by scanning 500 or more pollen grains per sample. CC (ChemCentre), marri (*Corymbia calophylla*), Jarrah (*Eucalyptus marginata*), Blackbutt (*Eucalyptus patens*), Myrt. (Myrtaceae).
(DOCX)

**S2 Table. Identification of proteins in 55 bee venom samples by LC-MS/MS.** Protein identification was confirmed by tryptic peptide match to the *Apis mellifera* reference proteome, requiring at least two unique peptides. (✔) Proteins previously reported in literature accounts of bee venom proteomics (Li et al., 2013; Matysiak et al, 2014; Matysiak et al, 2016). MW, Molecular weight of intact protein; kDa (Kilodalton).
(DOCX)

**S3 Table. The major contributing compositional proteins of bee venom.** A complementary list to Fig 3, presenting higher abundant proteins represented by the combined peptide match score (Sum PEP Score) and number of peptide spectrum matches (PSMs). MW: Molecular weight of intact protein; kDa (kilodalton).
(DOCX)

**S4 Table. List of the bee venom samples harvested in the field sites including analysed variables.** Bee venom weights and information on ecological and biological variables are presented (Temperature, Humidity, Nutritional supply, Historical hives' movement, Behavioural Flowering index, Nectar volume). Behavioural response: active (>80 bees reacting to the stimulating device); docile (<80 bees reacting to the stimulating device); Nutritional supply: PSS: unspecified pollen plus soy flour plus sugar syrup; MP: marri pollen; N: None; Flowering index (MFI; according to Campbell and Fearns, 2018).
(DOCX)

**S1 Appendix. Quantification of pollen in honey samples collected in the field sites.**
(DOCX)

## Acknowledgments

We acknowledge the planning, research and applicable science provided by ChemCentre and our industry partner, Australian Natural Biotechnology, and the assistance of Andrea Aromatisi for aiding the fieldwork. We are grateful to apiarists Peter Detchon and David & Leilani Leyland for hosting field sites and providing bee colonies.

## Author Contributions

**Conceptualization:** Daniela Scaccabarozzi, Kenneth Dods, Ben Pan Wafujian, Colin Priddis.

**Data curation:** Daniela Scaccabarozzi, Kenneth Dods, Thao T. Le, Joel P. A. Gummer, Michele Lussu, Colin Priddis.

**Formal analysis:** Joel P. A. Gummer, Michele Lussu.

**Funding acquisition:** Kenneth Dods.

**Investigation:** Daniela Scaccabarozzi, Kenneth Dods, Thao T. Le, Ben Pan Wafujian.

**Methodology:** Daniela Scaccabarozzi, Kenneth Dods, Thao T. Le, Joel P. A. Gummer, Michele Lussu, Lynne Milne, Tristan Campbell, Ben Pan Wafujian.

**Project administration:** Kenneth Dods, Colin Priddis.

**Resources:** Kenneth Dods, Tristan Campbell, Ben Pan Wafujian, Colin Priddis.

**Software:** Thao T. Le, Joel P. A. Gummer.

**Supervision:** Daniela Scaccabarozzi, Kenneth Dods, Ben Pan Wafujian, Colin Priddis.

**Validation:** Daniela Scaccabarozzi, Kenneth Dods, Michele Lussu, Colin Priddis.

**Visualization:** Daniela Scaccabarozzi, Kenneth Dods, Thao T. Le, Joel P. A. Gummer, Lynne Milne.

**Writing – original draft:** Daniela Scaccabarozzi.

**Writing – review & editing:** Daniela Scaccabarozzi, Kenneth Dods, Thao T. Le, Joel P. A. Gummer, Michele Lussu, Lynne Milne, Tristan Campbell, Ben Pan Wafujian, Colin Priddis.

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
