## [Decision Letter · Decision Letter 0]

20 Apr 2021

PONE-D-21-06677

Factors driving the compositional diversity of Bee Venom from a Corymbia calophylla (Marri) ecosystem, southwestern Australia

PLOS ONE

Dear Dr. Scaccabarozzi,

Thank you for submitting your manuscript to PLOS ONE. After careful consideration, we feel that it has merit but does not fully meet PLOS ONE’s publication criteria as it currently stands. Therefore, we invite you to submit a revised version of the manuscript that addresses the points raised during the review process.

We look forward to receiving your revised manuscript.

Kind regards,

Prof. Lorenzo Pecoraro

Academic Editor

PLOS ONE

Journal Requirements:

In your Methods section, please provide additional location information of the study sites, including geographic coordinates for the data set if available.

In your Methods section, please provide additional information regarding the permits you obtained for the work. Please ensure you have included the full name of the authority that approved the access to the honeybee colonies and, if no permits were required, a brief statement explaining why.

Thank you for stating the following in the Competing Interests section:

The authors have declared that no competing interests exist.

We note that one or more of the authors are employed by a commercial company: Australian Natural Biotechnology Pty. Ltd,

4a, Please provide an amended Funding Statement declaring this commercial affiliation, as well as a statement regarding the Role of Funders in your study. If the funding organization did not play a role in the study design, data collection and analysis, decision to publish, or preparation of the manuscript and only provided financial support in the form of authors' salaries and/or research materials, please review your statements relating to the author contributions, and ensure you have specifically and accurately indicated the role(s) that these authors had in your study. You can update author roles in the Author Contributions section of the online submission form.

4b,  Please also provide an updated Competing Interests Statement declaring this commercial affiliation along with any other relevant declarations relating to employment, consultancy, patents, products in development, or marketed products, etc. 

We note that Figure 1 in your submission contain map images which may be copyrighted. All PLOS content is published under the Creative Commons Attribution License (CC BY 4.0), which means that the manuscript, images, and Supporting Information files will be freely available online, and any third party is permitted to access, download, copy, distribute, and use these materials in any way, even commercially, with proper attribution. For these reasons, we cannot publish previously copyrighted maps or satellite images created using proprietary data, such as Google software (Google Maps, Street View, and Earth). For more information, see our copyright guidelines: http://journals.plos.org/plosone/s/licenses-and-copyright.

5a, You may seek permission from the original copyright holder of Figure 1 to publish the content specifically under the CC BY 4.0 license. 

5b, If you are unable to obtain permission from the original copyright holder to publish these figures under the CC BY 4.0 license or if the copyright holder’s requirements are incompatible with the CC BY 4.0 license, please either i) remove the figure or ii) supply a replacement figure that complies with the CC BY 4.0 license. Please check copyright information on all replacement figures and update the figure caption with source information. If applicable, please specify in the figure caption text when a figure is similar but not identical to the original image and is therefore for illustrative purposes only.

Reviewers' comments:

Reviewer's Responses to Questions

**Comments to the Author**

1. Is the manuscript technically sound, and do the data support the conclusions?

Reviewer #1: Yes

Reviewer #2: Yes

2. Has the statistical analysis been performed appropriately and rigorously? 

Reviewer #1: I Don't Know

Reviewer #2: Yes

3. Have the authors made all data underlying the findings in their manuscript fully available?

Reviewer #1: Yes

Reviewer #2: Yes

4. Is the manuscript presented in an intelligible fashion and written in standard English?

Reviewer #1: No

Reviewer #2: Yes

5. Review Comments to the Author

Reviewer #1: The manuscript is interesting and the integration of different approaches is appreciable. I have no major remarks but some minor issues.

- First of all the authors must revise the whole manuscript in terms of grammar and readability. Some sentences are quite odd or very hard to follow as they stand now. Please also check the typos in the presented figures

- Figures are really too much and most of them could be grouped in composite boxes ore moved to the supplementary information. Figure 3 is completely awkward and impossibile to read, whereas the PCA graphs seem quite unnecessary since no clear groups are visible.

ABSTRACT

line 19: please, provide a brief context to the applicative relevance of BV

Line 31-32: the sentence seems truncated. Please check.

Lines 36-37: Unclear sentence. Please rewrite.

Line 37: please, be clearer concerning the investigated behaviour.

INTRODUCTION:

line 112: which is the relevance of this plant in the context of the study?

METHODS:

lines 135-136: this sentence should be placed in the introduction

line 140: use the comma to delimit thousands.

line 144 fig.1: unclear sentence. Please rewrite

DISCUSSION:

line 370: please be clear about the issue of "certification". Is certification currently adopted for BV commercialization? Which kind of certification procedure are currently required for this product? Is there any international standard for certification of BV?

lines 373-377: this part seems an abstract/introduction. I would suggest to remove it from this chapter.

line 387: Did you measure the differences among colonies at the same site? Could this factor influence the site-specificity of BV yield and features?

CONCLUSION:

Apart from the future perspectives, this chapter looks like more as an abstract rather than a conclusion section. I would suggest to rewrite it.

Line 480: remove the word "fresh". It is not appropriate.

Reviewer #2: In this manuscipt the authors studied the influence of several ecological, management and behavioural factors in bee venom weight and protein composition. Bee venom (BV) is produced by honeybees and has been widely studied and increasingly sought for its therapeutic features and pharmaceutical applications. Several studies have explored the biological factors influencing bee venom protein nature and amount, but little research have analyzed the role of ecological factors. For this reason I think that this article is original, as based on an advanced proteomics technique combined with both ecological and biological information for data interpretation and prediction of causes behind BV variance. Authors provide an useful tool based on an extensive proteins’ assemblage of aid for establishing BV certification programs. This tool would also address the growing demand for BV in clinical and therapeutic fields, with substantial benefit to both human health and the beekeeping industry/primary producers.

The experimental plan is well organized, statistical analysis are appropriate and results are well argued. However, I believe that merging the results and discussion into a single section (Results and Discussion) would improve the quality of the article.

Moreover, I suggest the following minor revision:

- Figure 3 is not legible: use acronyms instead of whole names of the proteins, so as to avoid overlapping writings; moreover change the color of the bubbles to make writings visible;

- In figure 4 the legend is not necessary as the locations are shown in the base of the graph;

- Add the degree of significance in the boxplots in figures 4,7,9;

- In figures 5,6,8 check the percentages in axis 1 and 2 carefully. They are different from those reported in the main text;

- Lines 102-103: add the reference number for Ferreira et al (2010);

- Line 269: the authors writes “three target proteins”, but four proteins are listed between parentheses;

- Line 445: replace (Fig1) with (Fig 1).

- While revising your manuscript refer to the downloadable sample files to ensure that your submission meets the formatting requirements. References are not listed according to the guidelines of the Journal. Correct them.

- Please upload your figure files to the Preflight Analysis and Conversion Engine (PACE) digital diagnostic tool, http://pace.apexcovantage.com/. PACE helps ensure that figures meet PLOS requirements. To use PACE, you must first register as a user. Registration is free. Then, login and navigate to the UPLOAD tab, where you will find detailed instructions on how to use the tool.

6. PLOS authors have the option to publish the peer review history of their article (what does this mean?). If published, this will include your full peer review and any attached files.

Reviewer #1: No

Reviewer #2: No

---

## [Author Response · Author response to Decision Letter 0]

8 Jun 2021

We are grateful for the comments received by the editor and the reviewers. We refer to the cover letter and to the file 'responses to reviewers' for details.

---

## [Editor Report · Decision Letter 1]

15 Jun 2021

Factors driving the compositional diversity of Apis mellifera Bee Venom from a Corymbia calophylla (marri) ecosystem, Southwestern Australia

PONE-D-21-06677R1

Dear Dr. Scaccabarozzi,

We’re pleased to inform you that your manuscript has been judged scientifically suitable for publication and will be formally accepted for publication once it meets all outstanding technical requirements.

Kind regards,

Prof. Lorenzo Pecoraro

Academic Editor

PLOS ONE

Additional Editor Comments (optional):

All issues raised by the reviewers have been addressed in the revised manuscript, which is now suitable for publication. Congratulations!
---

## [Editor Report · Acceptance letter]

21 Jun 2021

PONE-D-21-06677R1 

Factors driving the compositional diversity of *Apis mellifera* Bee Venom from a *Corymbia calophylla* (marri) ecosystem, Southwestern Australia 

Dear Dr. Scaccabarozzi:

I'm pleased to inform you that your manuscript has been deemed suitable for publication in PLOS ONE. Congratulations! Your manuscript is now with our production department. 

Kind regards, 

on behalf of

Dr. Lorenzo Pecoraro 

Academic Editor

PLOS ONE